# Effects of Whey Peptides on the Quality of Pork Ball Preprepared Dishes during Repeated Freezing–Thawing

**DOI:** 10.3390/foods12193597

**Published:** 2023-09-27

**Authors:** Xiaotong Zhang, Pengjuan Yu, Jiayan Yan, Yishuang Shi, Jianhui Feng, Xinyan Peng

**Affiliations:** College of Life Sciences, Yantai University, Yantai 264005, China; zhangxiaotong0622@163.com (X.Z.); yupengjuan99@163.com (P.Y.); yanjiayan2023@163.com (J.Y.); shiyishuang2022@163.com (Y.S.); jhfeng0122@163.com (J.F.)

**Keywords:** repeated freeze–thaw, natural whey protein, FI, quality, pork ball preprepared dishes

## Abstract

This study evaluated the effect of FI (Fraction I, molecular weight < 1 kDa), which is separated from natural whey protein, on the antioxidant activity, sensory quality, color, texture characteristics, and microbial growth of pork balls during repeated freeze–thaw cycles (F-T cycles). The results indicated that pork balls mixed with FI significantly improved in quality after repeating the F-T cycle, especially with the addition of 10% FI. The quality was improved significantly after repeated F-T cycles by adding 10% FI, and the antioxidant activity after seven F-T cycles decreased by 40.78%, a similar result to that obtained with the addition of 0.02% BHA. In addition, FI effectively reduced the sensory damage of pork balls caused by repeated freezing–thawing and also significantly inhibited the growth of microorganisms. In summary, FI not only has excellent antioxidant capacity under repeated freeze–thaw conditions but also has significant antibacterial and quality preservation effects and is expected to be quantified as a kind of natural food additive with antibacterial and antioxidant properties. This paper not only explores the effect of FI on the quality characteristics of frozen and thawed pork balls in prepared dishes but also provides a theoretical basis for the application of whey polypeptides in prepared meat.

## 1. Introduction

With the onset of the fast food era, the popularity and consumption of prepared meat products such as pork meatballs, sausages, and patties have increased rapidly. Due to their convenience, satisfying sensory properties, and nutritional value, frozen pork balls have become popular and commonly consumed prepared meat products [1]. However, repeated freezing and thawing processes are frequently encountered during the process of production, processing, transportation, and storage, resulting in dehydration and oxidation deterioration, which affect product quality and have become limiting factors in the development of prefabricated meat dishes [2,3,4]. Therefore, finding effective techniques to improve the quality of prefabricated meat products throughout storage and processing is critical to encourage their continued development [5]. Meat shelf life is often described by its appearance, texture, color, flavor, microbiological activity, and nutritional content, and is also influenced by the freezing and thawing process [6]. Freezing is the most generally utilized long-term storage method for meat and meat products, as it enhances meat quality and increases shelf life by suppressing microbial development [7,8]. Temperature fluctuations inevitably occur during storage, production, sales, and long-distance transportation due to multiple transfers and imperfect cold chain conditions, as well as non-standard operations by sellers and consumers in daily life, resulting in the occurrence of the repeated freeze–thaw (F-T) phenomenon [9].

Natural whey protein (NWP), a byproduct of the casein or cheese-manufacturing industries, is a high-protein source that can be used as a food ingredient [10]. It is widely used in a variety of food applications due to its inexpensive cost, high nutritional content, and unique functional properties [11]. Whey protein contains several bioactive peptide fragments that can be released via enzyme hydrolysis (endogenous or exogenous hydrolysis), resulting in whey protein hydrolysate (WPH) [12]. WPH has been shown to have antioxidant, anti-inflammatory, antiobesity, and antihypertensive properties [13,14]. Among these, antioxidant activity is a significant biological property of whey peptides and has become a research focus for scientists [15,16]. Some peptides with antioxidant activity have been shown to have a high potential for usage as natural antioxidants in food items [17]. The addition of whey protein hydrolysate to a product has the potential to improve its stability and physicochemical qualities by reducing free radical production and suppressing oxidation. Peña-Ramos et al. [18] found that whey peptides added to cooked pork patties can effectively reduce cooking losses while also inhibiting lipid oxidation during freezing. Whey peptides have enabled significant advances in increasing the quality of meat products. According to the findings of Vavrusova et al. [19], whey peptides can significantly reduce high-valent heme iron while acting as effective iron chelators, free radical scavengers, and antioxidants. They act as dietary antioxidants, preventing oxidation during the storage and digestion of meat products.

Based on our previous studies [20], we discovered that whey polypeptide FI (molecular weight < 1 kDa) had the greatest reducing capacity. Adding FI to pork balls effectively regulated the oxidation reaction of the freeze–thaw meat system, limited water migration inside the meat mince, and had an important protective impact on meat quality. Therefore, the present study further explores the addition of the screened FIs to preformed pork meatballs based on the previous results to investigate the effects of their different concentrations on the quality of repeatedly frozen and thawed pork meatballs, thus further expanding the basis for the industrial application of whey peptides.

## 2. Materials and Methods

### 2.1. Chemicals and Materials

NWP (purity ≥ 90%) was obtained from Davisco Foods International, Inc. (Minnesota, MN, USA). FI (Fraction I, molecular weight < 1 kDa) was separated from the 4 h hydrolysates using 1 and 3 kDa ultrafiltration membranes according to Kong et al.’s method [12]. Fresh pork longissimus muscle, pork back fat, salt, soy sauce, rapeseed oil, pepper, nutmeg, monosodium glutamate, cooking wine, ginger, and spring onion were obtained from a local market (Yantai, Shandong, China). Alcalase (6 × 10^4^ U/g) was obtained from Novo Nordisk Biochem Inc. (Franklinton, NC, USA). Other chemicals were from Sinopharm Group Chemical Reagents Co., Ltd. (Shanghai, China). All reagents in this experiment were of analytical grade.

### 2.2. Preparation of Pork Balls

After the pork longissimus dorsi was peeled and the fascia was removed, the lean flesh was chopped into 3–5 cm pieces, minced with a meat grinder, and then added to a small amount of ice water. The formula excipients were calculated according to the weight percentage of the fresh pork (pork longissimus muscle: back fat was 7:3), including 20% ice water, 1% salt, 2% dark soy sauce, 15% rapeseed oil, 1.5% glucose, 0.2% pepper, 0.1% nutmeg, 0.5% monosodium glutamate, 1% cooking wine, 2% ginger, and 3% green onion. The control group received no additives, whereas the other five groups received 10% NWP, 5% FI, 10% FI, 15% FI, and 0.02% BHA. The following steps were undertaken: stir well, chop and mix, add remaining ice water to prevent protein denaturation, use a pill-rolling machine to make 3 cm pork balls, cook at 85 °C for 20 min, place in a tray wrapped in polyvinyl chloride film, and freeze in a −18 °C refrigerator. All ground pork was stored at −18 °C for 5 days and then thawed at 4 °C for 12 h until the central temperature reached from 0 to 2 °C, representing an F-T cycle, and freezing–thawing was repeated 3, 5, and 7 times.

### 2.3. Determination of Total Antioxidant Capacity

The total antioxidant capacity (TAC) of pork balls was determined using the method of Serpen et al. [21]. The absorbance of the sample was measured using ABTS+ radical cations (ABTS method). A concentration of 7 mmol/L ABTS was dissolved in water. ABTS radical cations (ABTS+) were produced by combining ABTS stock solution with 2.45 mmol/L potassium persulfate (final concentration) and storing the mixture in the dark for 12–16 h before use. Additionally, 10 mL of 0.1 mol/L phosphate buffered saline (PBS) was added to 2 g meat sample, which was then homogenized for 2 min before being centrifuged for 5 min at 4 °C at 1733× *g*. Using an AQ634 spectral analyzer (Yokogawa Co., Ltd., Shanghai, China) at 734 nm, the absorbance was determined by mixing 10 mL homogenate with 90 mL ABTS solution.

### 2.4. Determination of Pork Ball Color

A handheld CR400 spectrometer (Konica Minolta, Tokyo, Japan) was used to examine the color. The pork balls were cut into 3.0 cm × 3.0 cm × 1.0 cm cubes. The equipment was calibrated with a standard white plate before beginning the analysis. The data were recorded in L* (lightness from black to white), a* (redness to greenness), and b* (yellowness to blueness) color coordinates. The measurement was repeated three times, and the data were averaged.

### 2.5. Determination of Pork Ball Coliforms

We referred to Nikolaos et al.’s method and adjusted it [22]. First, 25 g of pork balls was accurately weighed and homogenized with 225 mL of sterile phosphate buffer at 11,260 × *g* for 2 min and then subjected to multiple 10-fold dilutions. Then, 1 mL of sample dilution within the dilution range from 10^−1^ to 10^−3^ was inoculated and mixed well with LST broth. This was followed by 24 h of incubation at 36 °C and looking for bubbles in an inverted tube. The culture was moved from the gas-producing LST broth tube to a brilliant green lactose bile (BGLB) tube and incubated for 48 h at 36 °C. Coliform-positive tubes were those that created gas. We performed three trials and computed the average. We obtained an MPN table and calculated the coliform MPN value per gram of sample.

### 2.6. Determination of Total Colonies

Total viable counts were evaluated using the method of Li et al. with modifications [23]. First, 25 g of meat sample was taken under sterile conditions, ground, dissolved in 225 mL of physiological saline, and diluted 10 times with a gradient concentration. A sterile plate was then filled with 1.0 mL of three consecutive dilution samples of different dilutions. They were then placed in a nutritional agar medium and incubated for 48 h at 37 °C for counting.

### 2.7. Determination of Texture Characteristics

Textural analysis of pork balls was conducted at room temperature (25 °C) using a TA-XT texture analyzer (TA-XT plus, Stable Micro Systems Ltd., Godalming, UK) and following the method of Cui et al. [24]. Firstly, the sample was cut into small pieces (20 mm × 20 mm × 10 mm). A cylindrical probe with a 50 N load cell and a P/36R flat-surface cylindrical probe was pressed vertically on top of the pork balls. Afterward, middle and back rates of 2.00 mm/s, 1.00 mm/s, and 1.00 mm/s were measured, and each sample was tested in triplicate. The test parameters were as follows: the measuring distance was 5 mm, and the trigger value was 5.0 N. Each batch was axially compressed twice to 30% of its original height in two consecutive cycles with a 10 s rest. Finally, the samples were analyzed with a texture analysis software (v1.18) program; the hardness, springiness, cohesiveness, chewiness, and other parameters were obtained, and the average value was given.

### 2.8. Sensory Evaluation

The descriptive aroma profile of Pan et al. [25] was used to evaluate the aroma profiles of meat samples. After thawing, the pork meatballs were heated to the core temperature of 85 °C, cooled to 20 °C, and cut into small pieces of 5.0 cm × 5.0 cm × 1.5 cm to evaluate the juiciness, racks, and overall acceptability of the pork meatballs. Each sample was encoded with a randomly selected three-digit number and provided to the evaluation team members in random order. Ten team members with experience in meat product evaluation were selected to use a 10-point scale to evaluate the sensory properties of cooked pork balls. The scores obtained were recorded on a table with a score of 1–10 (juicy, overall acceptability: 1 = extremely disliked, 10 = extremely liked; rancidity: 1 = no rancidity, 10 = severe rancidity). The sensory characteristics of pork balls with average scores of juiciness and overall acceptability between 6 and 10 with rancidity scores under 4 were considered acceptable. Between the evaluations of each sample, group members rinsed their mouths with warm water to avoid any impact. After completing each F-T cycle of the experiment, sensory analysis was conducted twice. The group’s average score for each sample was then calculated and analyzed.

### 2.9. Statistical Analysis

The effects of various dosage treatments on meat quality parameters were assessed using analysis of variance and a statistical analysis method. Statistix 8.1 software (Analytical Software, St. Paul, MN, USA) was used for all statistical analyses. We used analysis of variance (ANOVA) with Tukey’s multiple comparisons to measure the significance of the treatment effects (*p* < 0.05). The experiments were carried out three times with three replications. The data are shown as mean ± standard deviation values.

## 3. Results and Discussion

### 3.1. Effect of FI on the Antioxidant Activity of Freezing–Thawing Pork Balls

Antioxidant activity is a common food evaluation indicator that represents the ability to remove peroxides such as free radicals. Antioxidant activity can effectively reduce the oxidation of meat and even improve its composition, thus facilitating meat storage [26]. The changes in the antioxidant activity of pork balls with varied FI dosages during F-T cycles are displayed in Figure 1. The antioxidant capacity of the different treatment groups decreased significantly with the increase in freeze–thaw times (*p* < 0.05). The ABTS value in the control group was 0.15 mg/g before the freeze–thaw cycle, and it decreased by 53.33% after seven freeze–thaw cycles. The findings revealed that repeated freezing and thawing reduced antioxidant activity in meatballs. In comparison to the control group, varied doses of FI (5%, 10%, and 15%) and 0.02% BHA were found to increase the antioxidant capacity of pork balls. There was no significant difference between the antioxidant activity of the 0.02% BHA and the 10% FI groups, which were superior to that of the other groups. The findings showed that 10% FI could greatly increase the antioxidant activity of frozen and thawed pork balls in prepared foods. This may be explained by the whey peptides’ potent antioxidant activity [27], efficiently eliminating free radicals. This finding is consistent with Jimenez-Colmenero et al.’s study, which found that peptides influence the flavor of dry-cured ham and may have antioxidant properties [28]. Therefore, FI has the potential to take the place of artificial antioxidants.

### 3.2. Effects of FI on the Color of Pork Balls

The color of meat products influences consumer decisions significantly [29]. Figure 2 depicts the color difference results for pork meatballs with different dosages after numerous freeze–thaw cycles. As illustrated in Figure 2, for pork balls before the freeze–thaw cycle, the meatballs had the best color, exhibiting the highest values for L* and a* and the lowest value for b*. Because the NWP and FI powders themselves had a yellowish color, their L* values were lower than those of the control and BHA groups. The brightness of the pork balls gradually faded, and the L* decreased after seven freeze–thaw cycles. The cause of this phenomenon was related to muscle protein degeneration [30]. The control group showed the greatest decrease in L*, with a total decrease of 6.6. The 10% FI group experienced the smallest decline, with a 5.84 decrease in L* from 45.76 to 40.03. There are two possible explanations for these changes. Because the NWP and FI powders themselves had a yellowish color, their L* values were lower than those of the control and BHA groups. The brightness of the pork balls gradually faded, and the L* decreased after seven freeze–thaw cycles [30]. Second, iron and copper catalytic oxidation can cause color deepening and a drop in L* [31]. By efficiently preventing oxidation, 10% FI could induce a delayed drop in L*. 

As seen in Figure 2, when the number of F-T cycles grew, the a* value fell while the b* value increased. The main reason was that ice crystals pierced the cell membranes of pork balls during the freeze–thaw cycle, releasing pro-oxidants, and frozen storage could result in radical secondary lipid oxidation upon thawing, further deteriorating meatball quality in terms of flavor, color, odor, and nutritional value [32,33]. When the samples were compared after seven freeze–thaw cycles, the 10% FI group showed slower changes in the a* and b* values. As a result, 10% FI was more effective than other groups in delaying ice crystal growth and reducing lipid oxidation. Adding FI to repeatedly frozen and thawed meatballs has advantages in terms of maintaining the quality of meatballs.

### 3.3. Effects of FI on the Microbial Activity of Freezing–Thawing Pork Balls

Raw meat has an enriched nutritional profile, water activity from 0.98 to 0.99, and a pH range from 5.5 to 6.5, all of which encourage the growth of the majority of contaminating bacteria, raising the danger of contracting foodborne pathogens, as well as the prevalence of microorganisms that cause human disease [34]. Food manufacturers undertake product inspections that target indicator microorganisms such as coliform bacteria to accomplish hygienic control. As a result, coliform bacteria levels are regularly evaluated in meat, seafood, frozen desserts, soft drinks, and dairy products [35]. As a result, detecting coliforms in repeatedly frozen and thawed pork balls offers a more visualized reaction to microbiological contamination in frozen and thawed meat [36].

As shown in Table 1, the numbers of coliform bacteria in different additive groups in pork meatballs that underwent no freeze–thaw cycles differed slightly, and all of them were less than 3.00 MPN/g. After seven F-T cycles, the number of coliform bacteria in all groups increased significantly. The 0.02% BHA group saw little change, followed by the 10% FI group and the 15% FI group, which were 3.30 MPN/g and 3.20 MPN/g, respectively, which were significantly lower than the other treatment groups. Simultaneously, when combined with the total number of colonies analyzed in Figure 3, the total number of colonies in the control group was the largest (2.83 lg/CFU/g), with the lowest (2.07 lg/CFU/g) in 0.02% BHA, followed by 15% FI and 10% FI before the freeze–thaw cycles. The total number of colonies in each treatment group increased significantly (*p* < 0.05) after seven freeze–thaw cycles. This is consistent with the study of Mohammed et al. [37]. The total number of colonies in the control group increased to 5.72 lg/CFU/g. This may be attributed to ice crystal production and regeneration puncturing cells and causing the flow of cell contents, thus providing resources for microbial growth and reproduction [37]. At the same time, the slow thawing process’s low temperature provides time and circumstances for microbial multiplication. After seven freeze–thaw cycles, the colony growth of the 0.02% BHA group was effectively inhibited, and the total colony value was 3.91 lg/CFU/g. The 10% FI group and 15% FI group had an increase to 4.47 lg/CFU/g and 4.45 lg/CFU/g, respectively, and there was no significant difference (*p* > 0.05).

Rather than chemical preservatives, customers prefer natural food preservatives that are beneficial to their health [38]. FI is expected to be used in meat products, reduce microbial growth and reproduction, and delay meat deterioration.

### 3.4. Effect of FI on the Texture Characteristics of Freezing–Thawing Pork Balls

Texture is a significant indicator of meat product inspection [39] because it not only reflects the freshness of pork balls but also has a direct impact on their flavor and consumer preference. As shown in Figure 4, the quality of pork meatballs in all treatment groups dropped dramatically as the number of F-T cycles increased, notably after seven cycles. The fresh sample had a hardness of 187.86 N and a springiness of 0.85. The hardness and springiness of the control group samples decreased to 147.85 N and 0.39, respectively, after seven freeze–thaw cycles, which could be due to the fracture of myofibrillar caused by the growth of ice crystals during the process of multiple freeze–thaw cycles, resulting in muscle tissue and structural damage [40]. Furthermore, freezing-induced protein denaturation affects the tight topology of a three-dimensional network, reducing the hardness and springiness of the meatballs [41]. After seven F-T cycles, the hardness (162.90 N) and springiness (0.63) of pork ball samples in the 15% FI group had the smallest overall decrease, followed by the 10% FI group, with no significant difference (*p* > 0.05). The trends in cohesiveness, chewiness, hardness, and springiness were essentially the same. In conclusion, repeated freezing and thawing accelerated the destruction of myofibrillar integrity, resulting in the deterioration of texture properties. The addition of FI had a significant inhibitory effect on the decline in meatball quality, helping to minimize tissue damage and preserve food texture. The effects of 15% FI on the texture of pork meatballs were not significantly different from those of 10% FI (*p* > 0.05). Considering the economic benefits, adding 10% FI can achieve the optimal effect of protecting the quality of pork meatballs.

### 3.5. Effect of FI on the Sensory Characteristics of Freezing–Thawing Pork Balls 

For most customers, sensory qualities are the deciding element in food product selection [42]. The sensory qualities of pork balls during 0–7 freeze–thaw cycles are depicted in Figure 5. The pork meatballs in each treatment group had a good odor, rich juice, and high acceptance when the freeze–thaw cycles were not carried out. All of the samples’ sensory quality deteriorated as the number of freeze–thaw cycles increased. The amount of juice in the meat during chewing or mastication is referred to as juiciness, and it is one of the key meat qualities impacting eating quality [43]. The average score of succulence of meatballs in each group was greater than 6 points in the first five freeze–thaw cycles, which was within the acceptable range. The juice loss of pork meatballs was significant after seven F-T cycles, and the juiciness score did not reach the minimum acceptable value. Huang et al. found similar results, with dumplings with different amounts of fat added reducing in juiciness during freezing [43]. The average rancidity score of pork balls in each group increased gradually with the number of freeze–thaw cycles, which is similar to the findings of Choi et al. [44]. This conclusion could be attributed to the increased lipid oxidation in pork balls caused by freeze–thaw cycles, which promotes the development of rancidity odor [6,45]. Only the control and 10% NWP groups exhibited a rotten taste that exceeded the permissible range after seven cycles. These results indicated that the addition of FI and BHA effectively inhibited the production of a rancid taste in meatballs. After seven cycles, the trend for overall acceptance was consistent with juiciness, although only the control group was outside the acceptable range. However, if the meatballs were frozen and thawed no more than five times, the overall sensory quality wasaccepted. After five F-T cycles, the meatballs from the 10% FI group and the control group had a significant difference (*p* < 0.05) in sensory quality. This suggests that adding 10% FI to pork meatballs can effectively maintain sensory quality stability during the freeze–thaw cycle. Qin et al. [46] discovered that adding antioxidants to pork meatballs during storage could increase sensory quality while also extending shelf life. This could be due to the antioxidant activity of FI, and adding FI to pork balls acted as an antioxidant, delaying the oxidation reaction in the meatballs throughout the freeze–thaw process.

## 4. Conclusions

Food-derived bioactive peptides are highly valued for their broad range of activities. Incorporating bioactive peptides into meat can reduce protein oxidation, improve stability, and effectively slow down the quality deterioration caused by repeated freeze–thaw cycles in frozen meat and mince products. FI is made from natural whey protein that contains antioxidants. The antioxidant activity, sensory qualities, color, microbiological activity, and texture properties of pork meatballs were evaluated during the freeze–thaw process to determine the impacts of FI on their quality. After numerous freeze–thaw cycles, it was observed that the FI could successfully raise the oxidation stability, significantly minimize color deterioration, and greatly improve the sensory quality of pork meatballs. In conclusion, 10% FI can strengthen the protection influence on the quality of pork meatballs during repeated freeze–thaw cycles. This work provides a theoretical foundation for controlling oxidation and improving the quality of meat dishes using natural antioxidant polypeptides from whey protein.

## Figures and Tables

**Figure 1 foods-12-03597-f001:**
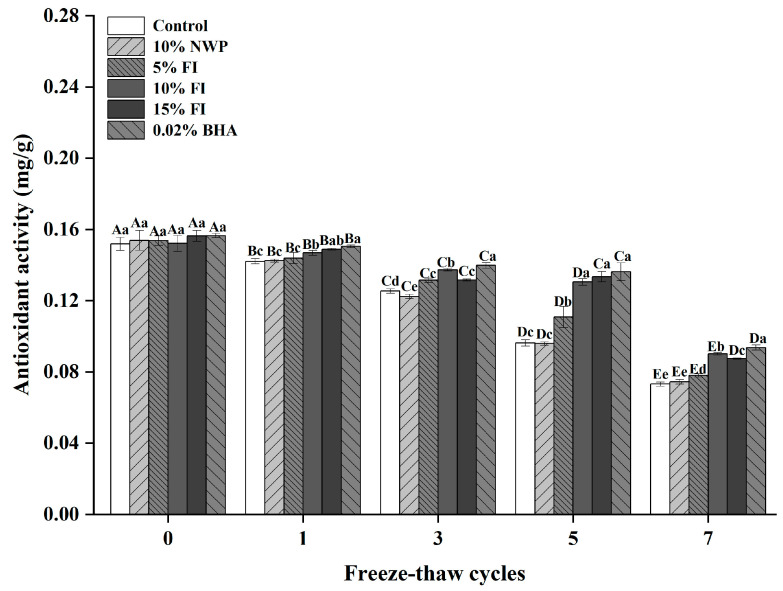
The antioxidant activity of pork balls with different FI contents during F-T cycles. Different uppercase letters (A–E) indicate significant differences between distinct cycles. Different lowercase letters (a–e) indicate significant differences between distinct samples. Control: without any additives in the sample; NWP: natural whey protein; BHA: butylated hydroxyanisole.

**Figure 2 foods-12-03597-f002:**
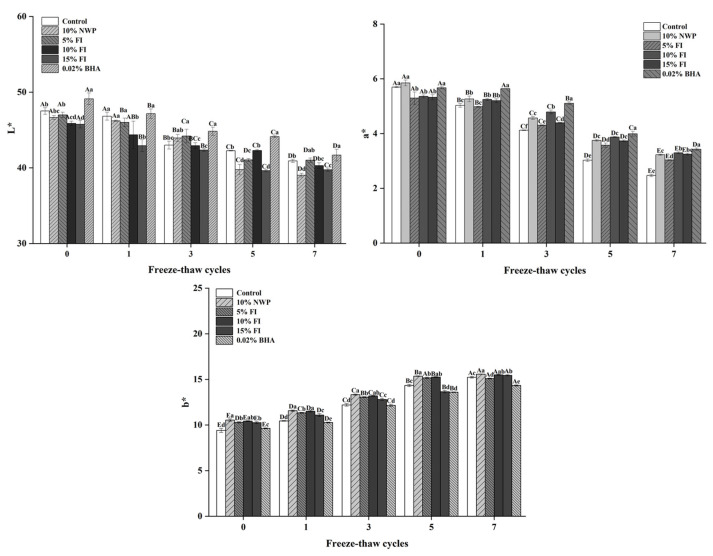
The color of pork balls with different FI contents during F-T cycles. Different uppercase letters (A–E) indicate significant differences between distinct cycles. Different lowercase letters (a–e) indicate significant differences between distinct samples. Control: without any additives in the sample; NWP: natural whey protein; BHA: butylated hydroxyanisole.

**Figure 3 foods-12-03597-f003:**
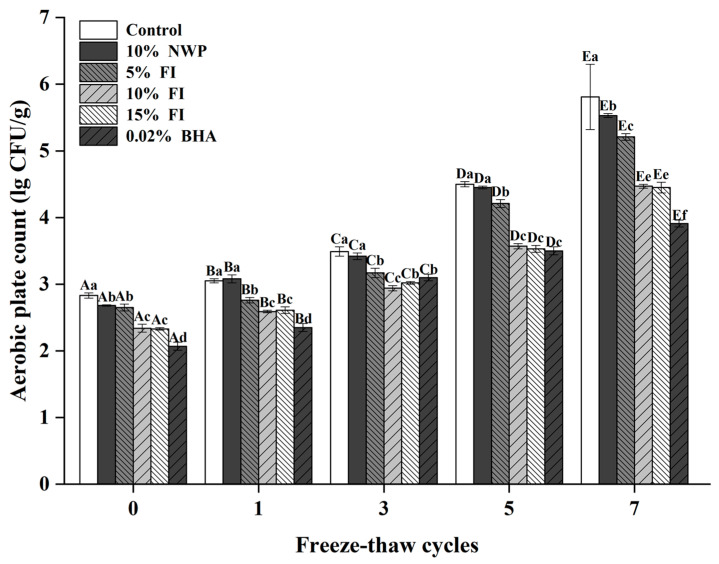
The aerobic plate count of pork balls with different FI contents during F-T cycles. Different uppercase letters (A–E) indicate significant differences between distinct cycles. Different lowercase letters (a–f) indicate significant differences between distinct samples. Control: without any additives in the sample; NWP: natural whey protein; BHA: butylated hydroxyanisole.

**Figure 4 foods-12-03597-f004:**
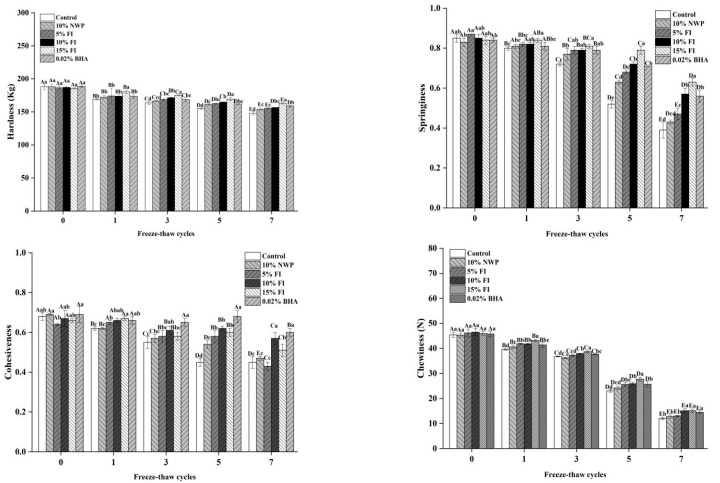
The textural properties of pork balls with different FI contents during F-T cycles. Different uppercase letters (A–E) indicate significant differences between distinct cycles. Different lowercase letters (a–e) indicate significant differences between distinct samples. Control: without any additives in the sample; NWP: natural whey protein; BHA: butylated hydroxyanisole.

**Figure 5 foods-12-03597-f005:**
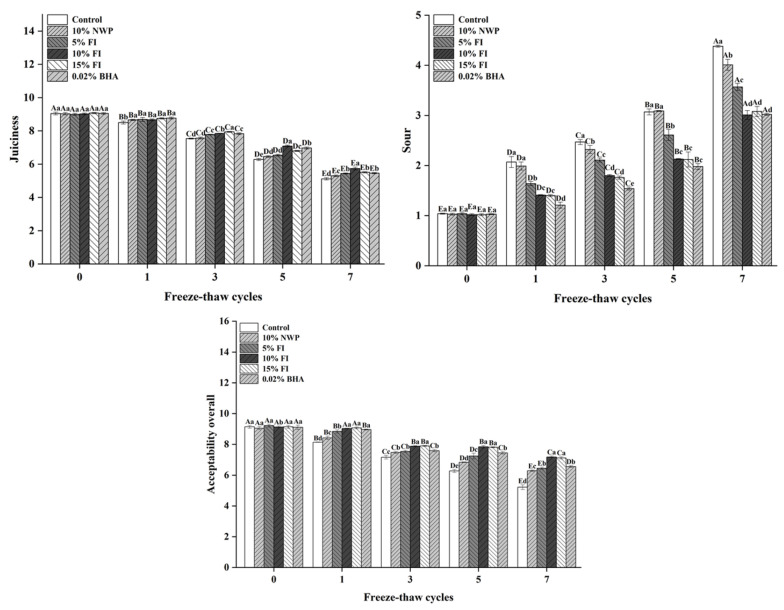
The sensory evaluation of pork balls with different FI contents during F-T cycles. Different uppercase letters (A–E) indicate significant differences between distinct cycles. Different lowercase letters (a–e) indicate significant differences between distinct samples. Control: without any additives in the sample; NWP: natural whey protein; BHA: butylated hydroxyanisole.

**Table 1 foods-12-03597-t001:** The *Coliforms* in MPN/g of pork balls with different FI additions during the F-T cycles. Control: without any additives in the sample; NWP: natural whey protein; BHA: butylated hydroxyanisole.

Group	Freeze–Thaw Cycles				
	0	1	3	5	7
Control	<3	<3	<3	3.6	7.2
10% NWP	<3	<3	<3	3.2	5.5
5% FI	<3	<3	<3	<3	3.7
10% FI	<3	<3	<3	<3	3.3
15% FI	<3	<3	<3	<3	3.2
0.02% BHA	<3	<3	<3	<3	<3

## Data Availability

The data presented in this study are available within the article.

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
