# Peer review of "Effects of Whey Peptides on the Quality of Pork Ball Preprepared Dishes during Repeated Freezing–Thawing"

_foods, 2023, doi:10.3390/foods12193597_

Round 1

Reviewer 1 Report

Comments and Suggestions for Authors

The paper from Zhang et al. addresses a very interesting topic. Increasing the quality of foodstuff, especially with by-products from other productions is gaining a lot of attention for food producers, consumers, and the scientific community. I find the manuscript worth of attention however I found it difficult to read in some parts.

From a general point of view, I found some sentences with the subject missing, commas used improperly, or even the use of "you" (see conclusions).

I think that the authors need to improve the general aspect and rephrase some sentences.  

Some specific comments:

Lines 30-33, 33-35, 51-52: add reference

Line  59-61, 82-85, 91-94: rephrase

Line 69-71: it seems that there is a different text format

Line 106: the sentence lacks of information about the instrument

Line 112-. Move 17 after method

Line 156: please correct

Comments on the Quality of English Language

From a general point of view, I found some sentences with the subject missing, commas used improperly, or even the use of "you" (see conclusions).

Please revise all along the manuscript.

Reviewer 2 Report

Comments and Suggestions for Authors

This work aims to demonstrate the beneficial effect of adding whey peptides in pork meat balls recipe when the prepared dish is repeatedly freeze and thaw. The Authors published at least three quite similar papers in which the same formulation of additives (10% NWP, 5% FI, 10% FI, 15% FI, or 0.02% BHA) was used on similar food products.

Some comments and requests

line 94 "repeated freeze-thaw.........please, specify the conditions adopted for thawing (time-temperature) and refreezing. How have you control the two processes?

line 120 please, specify the replicates of the microbiological analyses

line 188 please, define "high quality" of the not frozen pork balls color

line 194 "these changes can be explained in two explanations" please rewrite the sentence

line 240 Table 1 is not discussed or presented

line 250 for thee first time you introduce the term "springiness", even if this parameter was not listed among those reported in line 136-136

Comments on the Quality of English Language

I am not qualified to comment on English language

Reviewer 3 Report

Comments and Suggestions for Authors

General comments:

The description of materials and methods needs to be improved.

- Please include information on the origin of FI. It is a factor of the experiment.

- Please complete the information on thawing (e.g. method, temperature, time). This is key information regarding the experiment.

 The results and discussion are correct, but need to be improved and corrected (details below).

- In the description, do not repeat information on the level of p. Such information should be included in chapter 2.9.

- Please supplement with information on the total Delta-E color change of the product relative to the variant with BHA, as well as the change over the storage period.

- It is obvious that the drawings show "changes". Remove "changes" from the titles.

- Please verify the unit of strength in the graph showing hardness. The hardness is unusually large.

- Please verify the length of the vertical axes on the graphs. Springiness and cohesiveness do not take values above 1. Please make the vertical axes of the graphs real size.

Specific comments:

l.9 - Compare with l.49 - Fix the abbreviation for whey protein hydrolysate.

l.22 - "Ground pork" is redundant as it follows from other keywords.

l.103 - Instead of rpm, please specify times of earth acceleration (g).

l.103 - Please complete the information about the spectrometer (model, manufacturer).

l.106 - Please complete the information about the spectrometer (manufacturer).

l.107 - Necessary spaces before cm.

l.109 - Unnecessary double parenthesis.

l.119 - Necessarily explain the sentence in the context of Table 1 and Fig. 3.

l.125 - Necessary space before C.

l.128 - Please complete the information about the textrometer (model, manufacturer).

l.130 - From the description, it appears that a probe  plate of a certain diameter - compression platen (this is not a cylinder) - was used. Please specify the diameter instead of the area.

l.132 - What was the relaxation time between the first and second deformation.

l.133 - Trigger value is a force. Please verify the unit of measurement.

l.135 - Please verify the scope of the study. The discussion of "elasticity" and "other parameters" is missing from the rest of the manuscript. Instead of "cohesion" write "cohesiveness".

l.156 - Please clarify the meaning of p<0.05.

l.347,352,364,414 - Write the name of the journal in capital letters.

l.401 - Please verify the title. Strange characters in the title.
